# RouGE: Learning Gated Experts for Segment Anything in the Wild

## Abstract

Segment anything model (SAM) and its variants have recently shown promising performance as foundation models. However, existing SAM-based models can only handle scenarios seen during training, and usually suffer unstable performance when transferring to real-world unseen data, such as low-light, rainy or blurred images, which is crucial for applications such as autopilot. Therefore, adapting SAM-based models for real-world degradation while not impairing its original ability remains an open challenge. In this work, we propose a novel gated Mixture-of-Experts (MoE) structure, called RouGE, to improve the robustness of SAM-based models. Specifically, RouGE uses multiple lightweight probability gates to decompose complex real-world image conditions and judge whether the feature needs to be adjusted as well as to what extent the adjustment needs to be done, then handle them differently with a set of low-rank experts. During the inference stage, RouGE processes input images in a completely blind manner thus improving the model's performance in real-world scenarios. Extensive experiments demonstrate that RouGE consistently achieves state-of-the-art results on both degraded and clean images compared with other methods while tuning only 1.5% of parameters.

## 1 Introduction

Segment anything model (SAM) (Kirillov et al., 2023; Ravi et al., 2024) and its variants have recently shown impressive performance and have been widely applied in various downstream applications, *e.g.*, autopilot and medical image segmentation. However, existing SAM-based methods are usually trained on clean images without degradation. Given that degradations such as low light, rain and blur are almost unavoidable in real-world scenarios, existing models consequently suffer unstable performance when transferring to real-world unseen data. Therefore, how to improve the robustness of SAM-based models to deal with real-world diverse scenarios poses an open challenge.

To allow for a robust SAM-based model for real-world applications, several methods have been explored. For instance, one simple solution may use a two-step workflow with another image restoration model (Li et al., 2022; Wang et al., 2024a; Potlapalli et al., 2023) before segmentation to remove undesired degradations. Such methods rely heavily on the reconstruction results of pretrained image restoration models and cannot handle various types of degradation (*e.g.*, noise, blur or rain). More importantly, restored images may not benefit high-level visual tasks as they are initially designed for human eyes and may generate artifacts which has negative impacts on downstream tasks (Cui et al., 2021; Chen et al., 2024). To relieve such issues, other methods involve fine-tuning segmentation models that are tailored for specific degradation (Cui et al., 2021; Chen et al., 2023b). However, fine-tuning on specific degradation requires prior knowledge of the degradation type of input image, which is hard to achieve in real-world applications. Recently, Chen et al. (2024) proposed RobustSAM using a post-processing module to handle real degradation, but still suffers from heavy computational cost by increasing the model's parameters by about 32%.

Despite attempts have been made to obtain robust SAM models, current methods still face the following challenges. First, fine-tuning the foundation model can inherently degrade its original performance and lead to catastrophic forgetting problems. Second, the diversity of real-world degradation leads to significant variations in degradation types and a robust model needs to handle various degradation and clean images in a completely blind manner. Third, manually labeled real-world

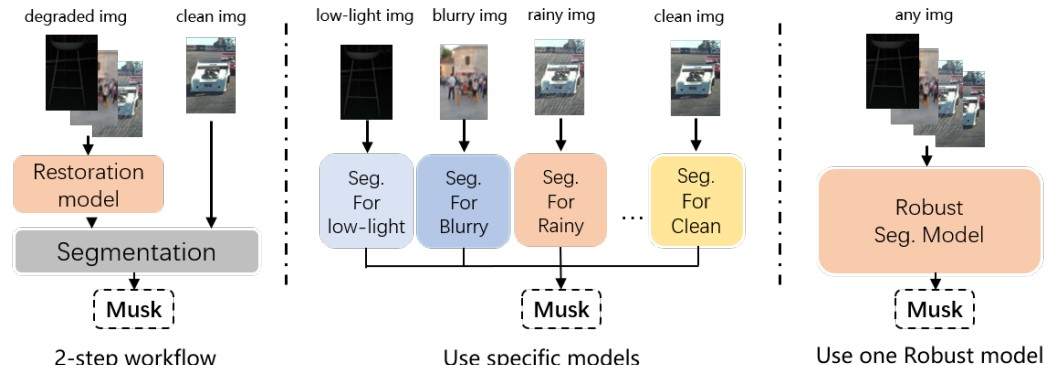

Figure 1: Comparison of Three Methods. Both the 2-step approach and using specialized models approach require obtaining prior knowledge of image categories for corresponding processing, and their workflow is complex. A robust model that can directly handle all types of images is evidently the optimal solution. Therefore, we propose a novel approach to empower a less-robust SAM-based model to become a robust model.

degradation data is scarce, and practical applications often require specific scene and degradation types, complicating model training further.

To maintain the original performance of the model, a good way is to minimize changes applied to the pretrained model weights with the Parameter Efficient Fine-Tuning (PEFT) techniques. By fine-tuning with almost all model parameters frozen, PEFT methods can adapt models to new domains while preserving the model's generalization ability at marginal cost and are widely used in both vision and natural language fields (Houlsby et al., 2019; Jie & Deng, 2022; Chen et al., 2022; Pfeiffer et al., 2020a; Wang et al., 2022; Yu et al., 2024). To tackle diverse data types, a natural approach is to break down complex tasks into multiple simpler tasks. By using mixture-of-expert (MoE)-like methods, we can decompose the complexity of real-world environments into multiple conditions, enabling us to use a set of smaller modules to solve the complex problem. Considering the third point, training with unlabeled images can better reduce the challenges of industrial applications.

Based on the above observations, we introduce a novel module **RouGE**, a plug-in **Ro**bustness-**U**plift module using **G**ated **E**xperts to perform differentiated processing on degraded and non-degraded inputs within a pretrained network. RouGE module comprises lightweight multiple probability gates and their corresponding low-rank experts (including lazy and trainable experts), to efficiently select suitable experts for input data and perform effective combined processing. The probability gates provide the model with interpretable classification capabilities to handle blind input, while the design of lazy and trainable experts endows the module with the ability to not disturb the distribution of the model's original parameters. Meanwhile, we propose an unsupervised imitation learning method designed for RouGE. We use unlabeled clean images to synthesize degraded images and let the model learn to narrow the gap between them. Through imitation learning, RouGE can be trained using a small amount of unlabeled images (approximately 2k images for each type of degradation), making it more suitable for industrial applications.

The main contributions of this work are summarized as follows: **(i)** We propose RouGE, a PEFT module designed for making pretrained less-robust SAM-based model a robust all-in-one model with marginal cost. The design of RouGE ensures the capability to maintain the model's original output features unchanged and only conduct selective feature modifications, thus avoiding the catastrophic forgetting issue associated with fine-tuning. **(ii)** We propose an unsupervised imitation learning approach, utilizing unlabeled images and synthesized degraded images for training, thereby circumventing the problem of missing labeled data and facilitating easier training of robust models for industrial applications. **(iii)** Our comprehensive experiments demonstrate that the RouGE method significantly enhances model robustness. Compared to the original model, RouGE can improve segmentation accuracy for degraded inputs by 4-13% in mAP with hardly any negative effects on results for non-degraded inputs. RouGE also outperforms other fine-tuning methods by a significant margin, even with a low trainable parameter ratio (about 1.56%).

## 2 RELATED WORKS

### 2.1 SAM AND ITS VARIANTS

Since the introduction of SAM, there has been a continuous emergence of derivative works (Zhang et al., 2023b; Zhang & Jiao, 2023). Researchers in the field of medical image segmentation are focused on fine-tuning SAM for high-quality medical image segmentation tasks (Zhang et al., 2023d; Mohapatra et al., 2023; de Oliveira et al., 2023; Li et al., 2024; Wu et al., 2023; Hu et al., 2023; Gao et al., 2023). In addition, there is a wealth of work fine-tuning SAM to adapt to other types of segmentation tasks such as satellite image segmentation (Ren et al., 2024), shadow Detection (Jie & Zhang, 2023; Chen et al., 2023c), marine animal segmentation (Zhang et al., 2024), and so on (He et al., 2024; Williams et al., 2023; Cao et al., 2023). Considering the vast parameter count of SAM, in real-world applications, compressed SAM models capable of real-time segmentation hold a higher value. Through knowledge distillation and model pruning, (Zhang et al., 2023a; Zhao et al., 2023; Xiong et al., 2023) have successfully compressed SAM models to a fraction of their original size. The EfficientSAM (Xiong et al., 2023) uses the MEA (He et al., 2022) method to distillate the pretrained SAM model and retains the performance of the SAM model most comprehensively. Recently, RobustSAM (Chen et al., 2024) has similarly noted the sensitivity of SAM to real-world degradation and used a post-processing module to handle this problem. However, their model still has a high learnable parameter count, which demands significant computational resources for both inference and training. By employing PEFT methods, we can effectively leverage the performance of the base model to achieve efficient model adjustments, thereby reducing computational costs.

### 2.2 PARAMETER EFFICIENT FINE-TUNING

The PEFT method has been widely applied to SAM-based models (Sahay & Savakis, 2024), primarily fall into prompt tuning (Wang et al., 2024b; Jia et al., 2022), adapter-like tuning (Houlsby et al., 2019; Jie & Deng, 2022; Chen et al., 2022; Wang et al., 2020; Pfeiffer et al., 2020a; Wang et al., 2022; Yu et al., 2024), partial tuning (Basu et al., 2024; Zaken et al., 2021) and reparameterization fine-tuning (Jie & Deng, 2023; Lian et al., 2022; Hu et al., 2021) categories. However, the primary application scenario of PEFT methods is fine-tuning models for downstream tasks and the original model's feature distribution would undergo significant disruption. To mitigate this issue, knowledge injection (Zhang et al., 2023e;c; Wang et al., 2020) and MoE-based (Shazeer et al., 2017; Kim et al., 2020; Yu et al., 2021) PEFT methods have been proposed. However, the former (Wang et al., 2020) requires explicit task annotations, while the latter (Wang et al., 2022; Chen et al., 2022) yields suboptimal results due to the lack of clear classification methods. These methods all fail to effectively enhance robustness.

## 3 METHODOLOGY

### 3.1 MOTIVATION

Segmentation models are sensitive to image degradation. When facing various types of degradation, SAM and SAM-based models may experience varying degrees of performance loss (Ji et al., 2024; Huang et al., 2023; Qiao et al., 2023; Wang et al., 2023). This is because degradation alters the overall or local distribution of image features, leading the model to erroneous perceptions. For instance, the texture of rainwater may cause black objects and shadows to be perceived as one entity, low-light environments may render object edges difficult to distinguish, and motion blur may lead to the cohesion of different objects. The SAM model exhibits relative robustness, whereas EfficientSAM performs comparatively worse. In Figure 2, we depict the segmentation performance degradation of SAM and the SAM-based EfficientSAM model when encountering three types of real-world degradation: low-light, motion

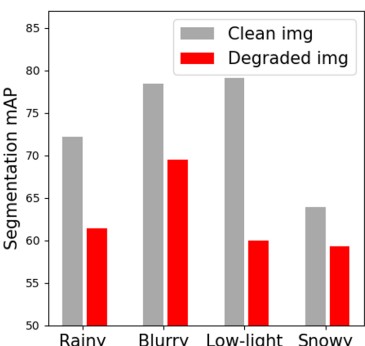

Figure 3: Impact of different type of real-world degradation on EfficientSAM

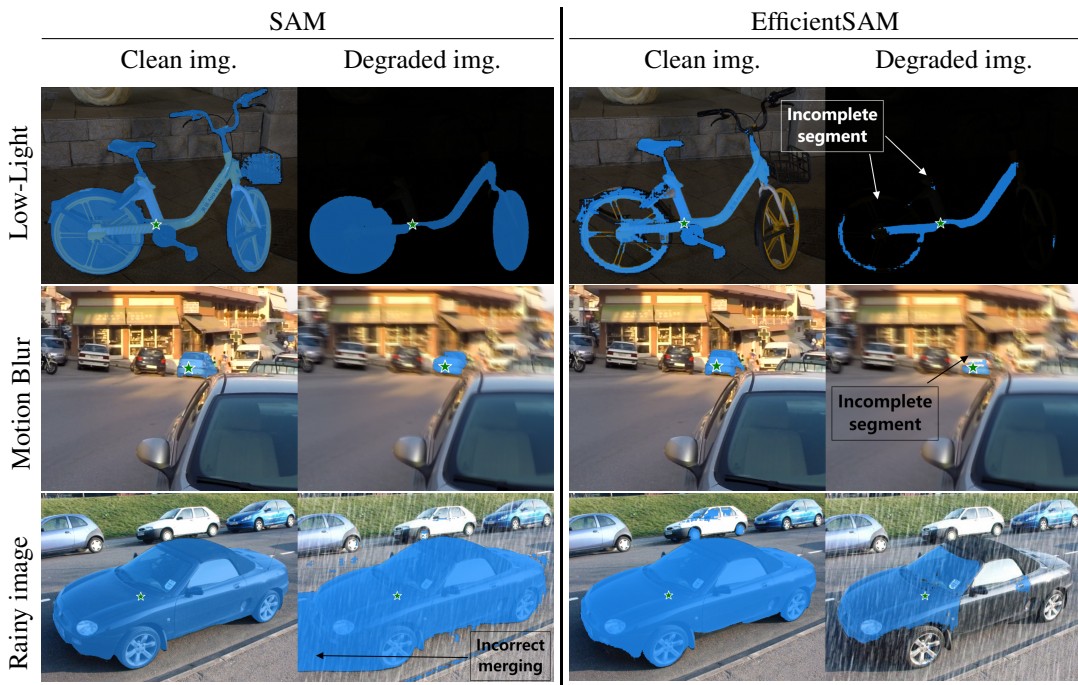

Figure 2: Presentation of the impact of various degradation on SAM and SAM-based Model

blur, and rainy conditions. How to mitigate the impact of real-world degradation on SAM-based models becomes an open challenge.

### 3.2 ROBUSTNESS-UPLIFT GATED EXPERTS

Fine-tuning pretrained models often leads to parameter drift, potentially resulting in models that only achieve domain adaptation rather than robustness improvement. Adapting the model to multiple domains simultaneously is the key point to achieving targeted robustness enhancement. Designs like that of Adapter-Hub (Poth et al., 2023; Pfeiffer et al., 2020b) can provide manual domain switching for models. From this, we conceive integrating multiple adapters into a single module, enabling the module to learn automatic switching, thereby achieving performance improvements across multiple domains. Therefore, we propose RouGE model and its unsupervised training process. RouGE utilizes lightweight probability gates within the module to control the weights of various experts, achieving diverse processing for different images by assigning corresponding expert proportions.

#### 3.2.1 OVERALL FRAMEWORK

The design of the RouGE model follows three key principles: (i) Automatically differentiate different inputs without type labels; (ii) The module should have the ability to "do nothing" to ensure that the original performance of the base model remains undisturbed; (iii) Utilize a minimal number of trainable parameters to ensure the module's parameter efficiency.

To achieve the above three objectives, we present a module with multiple independent probability gates, lazy expert (Expert 0), and trainable experts (Expert $i$), as shown in Figure 4. During both training and inference stages, RouGE takes image features $\mathcal{F}_t$ and input features $x_t$ as inputs. $\mathcal{F}_t$ are fed into the probability gates to obtain the proportions of each expert. $x_t$ are passed into each expert, and the predicted results generated by experts are multiplied by their respective proportions before being summed up and outputted. During the training process, synthetic image pairs of degraded and clean images are used, aiming to enhance model robustness by aligning the model predictions of degraded images with those of clean images. During the inference phase, arbitrary types of images can be used without the need to distinguish whether they are degraded images or not. Next, we will proceed to introduce each module separately.

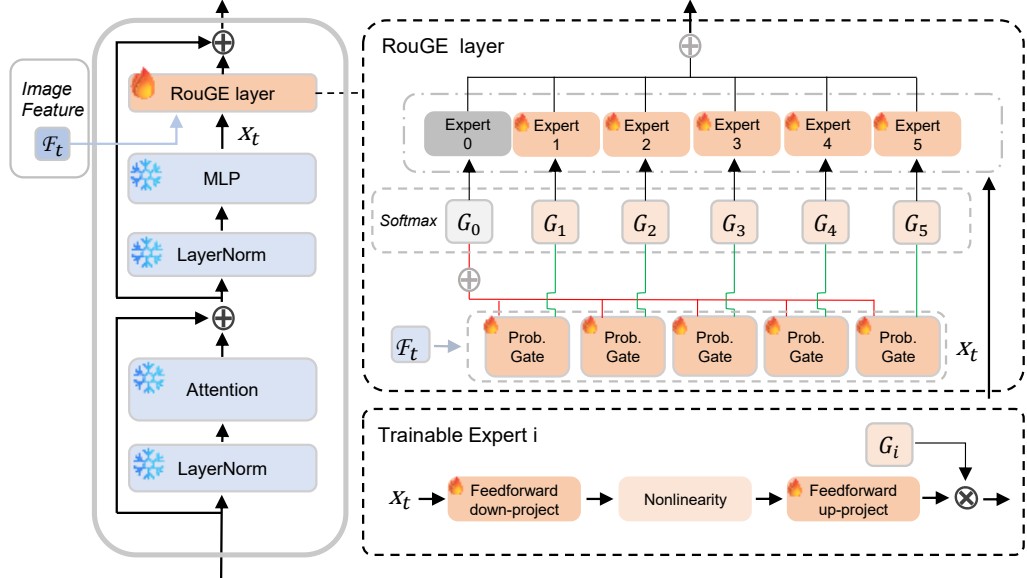

Figure 4: Structure of RouGE. RouGE module is inserted after the MLP layer of the transformer block, taking the MLP output features as input. The output result of the module is merged with the previous layer's features to obtain the output of the transformer block. This figure shows the structure of RouGE with 6 experts. Expert 0 is the lazy expert which directly multiplies the input data by $G_0$ and outputs the result. The other experts are trainable experts, composed of a dimension-reducing linear layer, a non-linear layer, and a dimension-restoring linear layer.

### 3.2.2 PROBABILITY GATES

The role of the probability gates is to distinguish between different types of input data and select the appropriate experts for processing. To reduce the noise in gate decisions, probability gates only take the image features $\mathcal{F}_t$ as input. In practical applications, we use global $\mathcal{F}_t$ extracted from the input image and the same $\mathcal{F}_t$ are used throughout the inference process for a single image. The probabilities generated by the probability gates are processed to serve as the output proportion parameters for each expert. The gate employs a lightweight structure of dual-layer fully connected layers, outputting a floating-point number between zero and one representing the acceptance probability.

To maintain the stability of the output features, the outputs of gates are concatenated and undergo a softmax function. In the case of having $n$ experts, the number of gates is $n - 1$. The acceptance probability of gate $i$ is $G_i$. The total rejection probability $G_0$ is obtained by summing up the rejection probabilities of all gates and dividing by the total number of gates. After undergoing softmax processing, the sum of all probabilities equals one, meaning the proportions of each expert sum up to one. So we define the output of the probability gates as probability vector $\mathbf{G}$.

$$\mathbf{G} = \text{Softmax}([\frac{1}{n-1}\sum_{i}^{n-1} 1 - G_i, G_1, G_2, ..., G_{n-1}]). \tag{1}$$

### 3.2.3 BOTTLENECK STRUCTURE EXPERTS

The probability vector $\mathbf{G}$ generated by the probability gates controls the weights of the experts and makes experts specializing in the current type of degradation exert their maximum impact. Each expert possesses its processing expertise after training and experts are divided into a lazy expert and multiple trainable experts. Lazy expert directly forwards input features to avoid introducing any bias and trainable experts introduce trainable parameters to fix different types of degradation.

To limit the number of trainable parameters in the model, we adopt the adapter (Houlsby et al., 2019)-like bottleneck structure, which includes a down-projection layer with parameters $\boldsymbol{W}_D \in \mathbb{R}^{m \times n}$ and an up-projection layer with parameters $\boldsymbol{W}_U \in \mathbb{R}^{n \times m}$. $m$ is the input dimension and $n$ is

the bottleneck middle dimension, with $n \ll m$ . The experts take $x_t$ as input and output $E_i(x_t)$ of the same size. Experts can be formulated as

$$E_i(x_t) = \begin{cases} \text{GeLU}(x_t \cdot \boldsymbol{W}_D^i) \cdot \boldsymbol{W}_U^i, & i \in [1, n-1] \\ x_t, & i = 0 \end{cases} \qquad (2)$$

Experts utilize a very small number of trainable parameters to ensure the efficiency of the module's parameters. We also compared the effectiveness of using other low-rank expert structures, as described in section 4.4. After being weighted by $\mathbf{G}$, the output $y_t$ of the module can be defined as

$$\mathbf{E} = [x_t, E_1(x_t), E_2(x_t), ..., E_{n-1}(x_t)], \qquad (3)$$

$$y_t = \mathbf{G}\mathbf{E}^T. \qquad (4)$$

Parameters of all experts and probability gates are updated during the entire training stage. We do not manually specify experts for each type of input data. Instead, we allow the model to finely decompose task types, and use combinations of multiple experts to achieve better processing results through fully end-to-end training. By analyzing $\mathbf{G}$, we show it in section 4.3.2 that the trained RouGE model can differentiate between different types of input data and process them in a targeted manner. Additionally, we experimentally validated the effectiveness of the lazy expert in section 4.5.

Through ablation studies, we conclude that RouGE does not need to be added to every transformer block. Instead, adding them only to the final few blocks of the model can achieve better results and more details can be found in appendix A.1.

### 3.3 Loss Function for Unsupervised Imitation Learning

To better train the lightweight probability gate and experts and mitigate the absence of labeled degraded image data, we employ a method based on imitation learning to minimize noise during training. In the field of image restoration, a significant amount of artificially synthesized degraded datasets (such as rainy or foggy images) are proposed for restoration models. These datasets include pairs of clean images and degraded images generated by adding specific types of degradation. Our training method precisely leverages these datasets to teach the model how to "ignore" these degradations. Since the content in the images is consistent, the segmentation of the same object should yield identical ground truth results.

During training, we utilize models that do not include RouGE ( $\mathcal{M}_{ori}$) and use clean images $I_{clean}$ as input to obtain clean outputs $\mathcal{M}_{ori}(I_{clean})$ as targets $T$.

$$T = Mask(\mathcal{M}_{ori}(I_{clean})). \qquad (5)$$

Next, we feed both clean images $I_{clean}$ and degraded images $I_{deg}$ into models containing RouGE ( $\mathcal{M}_{RouGE}$), respectively. We then compute losses by comparing the results with $T$ separately and perform backpropagation. The training objective is to ensure that models containing RouGE produce consistently high-quality results when faced with a set of clean images and degraded images with the same content. In the segment anything task, we employed the combination of Dice Loss and Sigmoid Focal Loss as the loss function $Loss_{dice\&focal}$. The training loss can be described as

$$L_{clean} = Loss_{dice\&focal}(T, \mathcal{M}_{RouGE}(I_{clean})), \qquad (6)$$

$$L_{deg} = Loss_{dice\&focal}(T, \mathcal{M}_{RouGE}(I_{deg})). \qquad (7)$$

## 4 Experiments

We evaluated the effectiveness of RouGE on image segmentation tasks. First, we introduce the experimental settings in section 4.1, covering the use of datasets, backbone selection, and the settings of other baseline methods. In section 4.2, we compare RouGE with other baseline models and provide a comprehensive analysis of the results. Next, in section 4.3, we empirically validate the automatic classification capability and out-of-domain performance of RouGE, and also compare the performance of restore-then-segment with RouGE. Finally, in section 4.4, we conduct other ablation experiments to explain its superiority.

## 4.1 Experimental Settings

**Datasets.** We obtained the Rain200L (Yang et al., 2017), DDN (Fu et al., 2017), GoPro (Nah et al., 2017), LIS (Chen et al., 2023a), and Snow100k (Liu et al., 2018) datasets from the image restoration domain and split them to serve as training and testing data for the model. Additionally, we used CityRain and CityFoggy (Cordts et al., 2015; 2016), as additional test data for experimentation. Among these, Rain200L and DDN consist of rainy weathered image pairs created using different methods, GoPro comprises dynamic blurry image pairs, LIS includes low-light image pairs, and Snow100k consists of snowy weathered image pairs.

All these datasets include artificially synthesized degraded images of varying degrees as well as their original clean images. Since the segment anything task requires a point prompt or bounding box as input, we selected the clean images from all image pairs and obtained bounding boxes in the images using a state-of-the-art object detection model. Subsequently, we inputted the bounding boxes and clean images into the Segment Anything model (Kirillov et al., 2023) to obtain the ground truth masks for the quantitative test. To avoid selecting points that are off-center from the object, we performed an erosion operation on the ground truth mask and then randomly selected a point as the point prompt. A set of image pairs uses the same point prompt and ground truth mask because the non-noise information on the image pairs is identical. We utilized CLIP's image encoder (Radford et al., 2021) as the image feature extractor in the experiment and pre-extracted image features for each in. The ablation study on image feature selection is in Appendix 10.

**Pretrained backbone.** We adopt the EfficientSAM-Ti (Xiong et al., 2023) as the backbone model and we utilized the pretrained parameters provided by the authors of EfficientSAM. The model comprises a transformer-based image encoder and a mask decoder. It takes the input image and point prompt and outputs the mask of the object pointed to by that point on the image.

**baseline models.** We selected 8 baseline models from 4 categories of methods for comparative experiments and the replicated RobustSAM on the EfficientSAM-Ti. Categorized by type, we selected (i) full fine-tuning. (ii) adapter-based: Adapter (Houlsby et al., 2019), Convpass (Jie & Deng, 2022), Adaptformer (Chen et al., 2022). (iii) partial fine-tuning: LN (Basu et al., 2024), Bitfit (Zaken et al., 2021). (iv) mixture-of-adapter: Adamix (Wang et al., 2022), AdapterFusion (Pfeiffer et al., 2020a) (v) RobustSAM (Chen et al., 2024): employing AMFG-F, AOTG, and ROT on base model, as our baseline methods. Additionally, while the RobustSAM paper employed supervised learning, we opted for unsupervised learning to ensure fairness.

**Hyperparameter settings.** In all experiments, we use AdamW as the optimizer with $lr = 1e - 4, weightdecay = 5e - 2$. We use the combination of dice loss and sigmoid focal loss as the loss function and each accounts for 50%. In the absence of specific instructions, we set the number of experts in the RouGE model to 6. Additionally, the RouGE model is only added to the last two layers of the transformer block in the image encoder. During training, we utilized a NVIDIA GeForce RTX 3090 GPU and the five datasets were alternated sequentially to train a robust model capable of handling five types of degradation simultaneously.

## 4.2 Comparison with SOTA methods

### 4.2.1 Quantitative experiment

We performed unsupervised training on the Rain200L, DDN, GoPro, LIS, and Snow100k datasets simultaneously and compared different fine-tuning methods, as shown in Table 1. As can be observed, our proposed RouGE method demonstrates strong robustness improvement capabilities. RouGE can significantly enhance the model's mean average precision on both degraded and non-degraded data. Compared to other methods, the RouGE model exhibits the greatest improvement in mAP and reduces the performance degradation between degraded and non-degraded data. From the table, it can be observed that other methods fail to balance the segmentation performance between degraded and non-degraded inputs, thus validating our argument. Moreover, the full fine-tuning method performs poorly when the available dataset size is small. The trainable parameters of the RouGE model account for only 1.56% of the total fine-tuning parameters. In summary, the RouGE model surpasses other methods, achieving state-of-the-art performance in enhancing robustness.

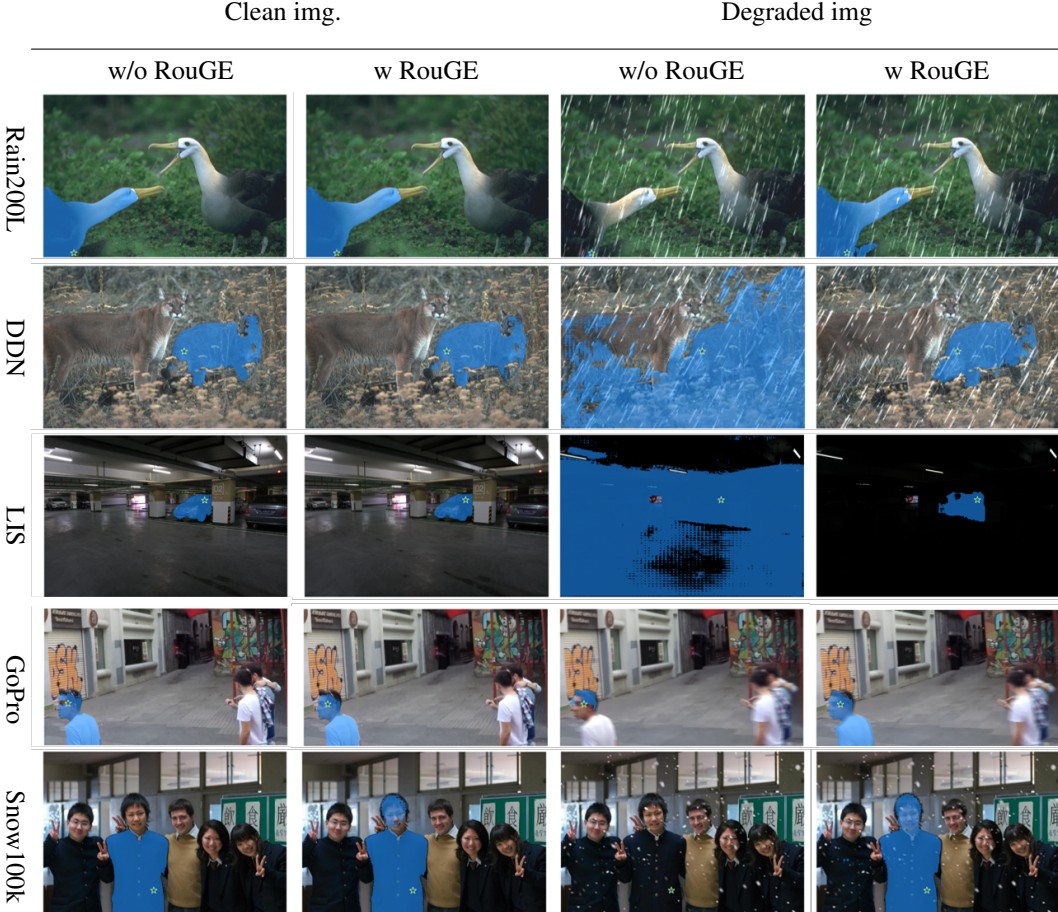

Figure 5: Presentation of the Segmentation performance

Table 1: **Comparison with other methods on Robustness-uplift benchmark.** We report the mean average precision (mAP) of various methods on the test set. **Bold** number indicates the best value for that data type.

| Method | Params | Rain200L Clean | Rain200L Rainy | DDN Clean | DDN Rainy | GoPro Clean | GoPro Blurry | LIS Clean | LIS Low-light | Snow100k Clean | Snow100k Snowy |
|---|---|---|---|---|---|---|---|---|---|---|---|
| Base model | - | 74.13 | 72.82 | 72.23 | 61.40 | 78.48 | 69.48 | **79.12** | 60.02 | 63.92 | 59.37 |
| Full fine-tune | 100% | 69.57 | 67.85 | 65.09 | 62.52 | 61.05 | 59.55 | 64.60 | 59.45 | 58.85 | 59.98 |
| LN | 0.18% | 75.44 | 74.48 | 72.9 | 69.05 | 78.62 | 71.73 | 78.71 | 65.47 | 64.02 | 62.51 |
| Bitfit | 0.51% | 75.95 | 74.15 | 73.43 | 65.99 | 73.84 | 69.03 | 78.29 | 63.17 | 63.17 | 61.74 |
| Adapter | 1.47% | 74.19 | 74.96 | 75.94 | 69.78 | 76.93 | 71.51 | 77.72 | 64.09 | 63.75 | 62.08 |
| Convpass | 2.00% | 74.50 | 73.52 | 72.06 | 60.11 | 73.85 | 70.34 | 69.21 | 62.01 | 63.80 | 61.43 |
| Adaptformer | 8.61% | 74.58 | 75.01 | 75.58 | 70.76 | 76.52 | 71.66 | 77.19 | 64.93 | 63.98 | 62.55 |
| Adamix | 3.67% | 72.38 | 71.02 | 75.73 | 63.96 | 79.85 | 75.22 | 75.96 | 63.85 | 63.72 | 60.91 |
| AdapterFusion | 7.47% | 73.92 | 72.67 | 73.71 | 71.61 | 77.72 | 70.61 | 73.19 | 64.45 | 63.62 | 60.09 |
| RobustSAM | 32.11% | 75.66 | 75.82 | 77.47 | 73.33 | 79.75 | **75.74** | 77.06 | 65.20 | 65.00 | **63.19** |
| RouGE (Ours) | 1.56% | **76.61** | **76.99** | **77.75** | **74.85** | **80.01** | 75.17 | 78.77 | **65.62** | **65.34** | 63.14 |

### 4.2.2 QUALITATIVE EXPERIMENT

In Figure 5, we show the robustness enhancement ability of RouGE. For clean image inputs, the insertion of RouGE only brings minimal changes in segmentation outcomes. Conversely, for inputs with various types of degradation, the insertion of RouGE significantly improved model segmentation results. This aligns with our previously conducted quantitative experiments. More segmentation results can be found in appendix A.4

Table 2: Restore-then-Segment experiments. We report the mean average precision (mAP). **Bold** indicates better result.

| Method | Rain200L-Rainy | DDN-Rainy | GoPro-Blurry | LIS-Low-light | Snow100k-Snowy |
|---|---|---|---|---|---|
| Base model | 72.82 | 61.40 | 69.48 | 60.02 | 59.37 |
| AirNet+Base model | 73.56 | 66.31 | 69.94 | 53.53 | 60.18 |
| RouGE+Base model | **76.99** | **74.85** | **75.17** | **65.62** | **63.14** |

## 4.3 DISCUSSION

### 4.3.1 RESTORE THEN SEGMENT?

The goal of image restoration tasks is to restore images to a form that is more friendly to the human eye instead of downstream visual tasks. We used AirNet (Li et al., 2022) as the image restoration model and conducted segmentation experiments on degraded images after restoration. From Table 2, we can see that the improvement of image restoration on downstream segmentation tasks is limited.

### 4.3.2 TASK DISCRIMINATION CAPABILITY OF ROUGE

RouGE model utilizes probability gates to distinguish input data, thereby achieving automatic classification and processing of unlabeled data. The outputs of these gates demonstrate strong interpretability and can be utilized for feature analysis. By using the output of the probability gates in the RouGE model as features for t-SNE clustering, we obtained a visualization that demonstrates how the RouGE model classifies and processes the input. From the clustering results, it is evident that there is a clear distinction between degraded and non-degraded data. Moreover, due to varying degrees of degradation, some degraded data may bear similarities to non-degraded data, while others exhibit significant differences.

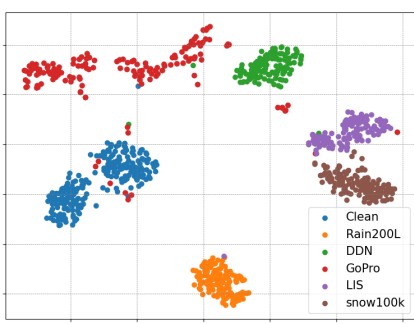

Figure 6: t-SNE visualization of probability vector $G$.

### 4.3.3 OUT-OF-DOMAIN EXPERIMENTS

To further demonstrate that the RouGE model does not disrupt the original model's parameter distribution, we conducted out-of-domain data experiments. We conducted tests using the CityRain and CityFoggy datasets, which contain road images under normal weather conditions as well as rainy and foggy weather conditions. We compared the segmentation performance of the RouGE model with that of the base model.

Table 3: Out-of-domain experiments. We report the mean average precision (mAP). **Bold** indicates better result.

| Method | Cityrain | | Cityfoggy | |
|---|---|---|---|---|
| | Clean | Rainy | Clean | Foggy |
| Base model | 69.49 | 61.23 | 71.21 | 56.14 |
| RouGE | **70.27** | **66.60** | **75.38** | **71.29** |

From Table 3, the RouGE model exhibits equally outstanding performance in out-of-domain scenarios, preserving the model's original performance intact. Moreover, it demonstrates performance improvements for similar degradation types like rainy images(Cityrain) and exhibits commendable zero-shot performance for unseen degradation types like foggy images(Cityfoggy).

## 4.4 ABLATION STUDIES

### 4.4.1 EXPERT DESIGN

In addition to Adapter-like experts, we also explored the effectiveness of other types of experts. In previous experiments, we found that the LN method, which fine-tunes the affine transformation parameters of the LN layers, could achieve relatively good results. Therefore, we considered testing the use of affine-based experts. The affine expert we designed contains a set of affine transform

Table 4: Test the performance of Adapter-like expert and Affine-based expert.

| Method | Rain200L | | DDN | | GoPro | | LIS | | Snow100k | |
|---|---|---|---|---|---|---|---|---|---|---|
| | Clean | Rainy | Clean | Rainy | Clean | Blurry | Clean | Low-light | Clean | Snowy |
| Affine expert | 74.86 | 75.57 | 75.28 | 70.68 | **80.26** | 73.75 | **78.89** | 65.57 | **65.49** | **63.26** |
| Adapter expert | **76.61** | **76.99** | **77.75** | **74.85** | 80.01 | **75.17** | 78.77 | **65.62** | 65.34 | 63.14 |

Table 5: Ablation study on lazy expert

| Method | Rain200L | | DDN | | GoPro | | LIS | | Snow100k | |
|---|---|---|---|---|---|---|---|---|---|---|
| | Clean | Rainy | Clean | Rainy | Clean | Blurry | Clean | Low-light | Clean | Snowy |
| w/o lazy expert | 75.76 | **77.27** | 76.04 | 71.96 | 79.06 | **75.24** | 77.64 | 64.46 | 63.98 | **63.43** |
| w lazy expert | **76.61** | 76.99 | **77.75** | **74.85** | **80.01** | 75.17 | **78.77** | **65.62** | **65.34** | 63.14 |

Table 6: Comparison between supervised and unsupervised RouGE model

| Method | Rain200L | | DDN | | GoPro | | LIS | | Snow100k | |
|---|---|---|---|---|---|---|---|---|---|---|
| | Clean | Rainy | Clean | Rainy | Clean | Blurry | Clean | Low-light | Clean | Snowy |
| Base model | 74.13 | 72.82 | 72.23 | 61.40 | 78.48 | 69.48 | 79.12 | 60.02 | 63.92 | 59.37 |
| Unsupervised | 75.44 | 74.48 | 72.9 | 69.05 | 78.62 | 71.73 | 78.91 | 65.47 | 64.02 | 62.51 |
| Supervised | **77.08** | **84.10** | **77.60** | **78.62** | **84.39** | **84.41** | **80.16** | **76.21** | **67.39** | **67.56** |

parameters, $\gamma$ and $\beta$, similar to the trainable parameters of the layer normalization layer. The comparative experimental results are presented in Table 4. Based on the results, the performance of the Adapter-like expert is better than that of the Affine-based expert.

## 4.5 LAZY EXPERT

The presence of lazy experts facilitates the model's ability to handle clean images and reduces training complexity. In the experiments, we maintained the same number of trainable experts. The results in Table 5 reflect the positive effects brought by the lazy expert.

### 4.5.1 SUPERVISED VS. UNSUPERVISED LEARNING

The unsupervised learning training of the RouGE model greatly reduces the cost of data acquisition. However, at the same time, unsupervised learning also makes RouGE relatively ineffective when faced with severely degraded types of data such as the LIS dataset. Using more accurate data labels for supervised learning can further enhance the model's capabilities. Based on unsupervised imitation learning and switching to using ground truth labels to train degraded images instead, We conducted a comparative experiment with supervised learning using labels generated by SAM, as shown in Table 6.

As observed, the model's detection accuracy on degraded images has significantly improved. Therefore, under circumstances where obtaining data labels is feasible, you can weigh the cost and benefit of obtaining labels to choose a more suitable training method.

## 5 CONCLUSION

In this paper, we propose a plugin robustness enhancement module, RouGE, which can enhance the robustness of pretrained SAM-based models at marginal cost. Experiments conducted both within and outside the domain demonstrate RouGE's capability to selectively modify degraded images while preserving the original performance of the model for clean images. Compared with existing PEFT methods and reproduction of RobustSAM, RouGE demonstrates superiority in both robustness enhancement capability and efficiency in terms of trainable parameters. RouGE model exhibits high versatility, as it can be seamlessly integrated into any transformer block. This renders it with the potential to be applied across various types of visual models. In the future, we will continue to explore the application of RouGE in a broader range of visual models and tasks.

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

## A  APPENDIX

### A.1  HOW MANY EXPERTS AND ROUGES ARE NEEDED?

The number of experts and the number of RouGE directly affect the trainable parameter count. Usually, fine-tuned models of like adapter are inserted into each transformer block. However, the RouGE model is more of a widening rather than a deepening model, so inserting it into all transformer blocks may not necessarily be advantageous. The base model's has 11 transformer layers, and we experimented with inserting the RouGE model starting from $i$th layer, as shown in 7. Training was conducted for 10 epochs in all cases. The experimental results indicate that the RouGE model is not suitable for being added to all blocks but rather for being added to the final few layers. In our previous experiments, we used the optimal configuration, adding RouGE to the last two layers.

For the number of experts, we conducted a comparative experiment as shown in 8.When the number of experts degrades to 2, the RouGE model becomes an adapter that can adjust fusion coefficients. As the number of experts increases, the effectiveness of RouGE improves. After balancing the parameter count and effectiveness, we chose $N = 6$ as the experimental hyperparameters setting.

Table 7: Inserting RouGE from $i$th layer.

| $i$ | Rain200L | | DDN | | GoPro | | LIS | | Snow100k | |
|---|---|---|---|---|---|---|---|---|---|---|
| | Clean | Rainy | Clean | Rainy | Clean | Blurry | Clean | Low-light | Clean | Snowy |
| $i = 0$ | 76.04 | 73.73 | 73.21 | 71.02 | 79.09 | **75.31** | 78.28 | 62.71 | 65.05 | 63.46 |
| $i = 5$ | 75.65 | 76.46 | 76.50 | 72.55 | 79.40 | 73.95 | 78.34 | 69.07 | **65.82** | **63.83** |
| $i = 9$ | 75.91 | 75.03 | 77.34 | 74.02 | 79.41 | 75.13 | 78.23 | 64.98 | 65.41 | 63.17 |
| $i = 10$ | **76.61** | **76.99** | **77.75** | **74.85** | **80.01** | 75.17 | **78.77** | **65.62** | 65.34 | 63.14 |
| $i = 11$ | 75.49 | 77.30 | 73.51 | 69.93 | 78.99 | 73.01 | 78.35 | 65.08 | 65.11 | 63.38 |

Table 8: RouGE with $N$ experts

| $N$ | Rain200L | | DDN | | GoPro | | LIS | | Snow100k | |
|---|---|---|---|---|---|---|---|---|---|---|
| | Clean | Rainy | Clean | Rainy | Clean | Blurry | Clean | Low-light | Clean | Snowy |
| $N = 2$ | 74.99 | 74.97 | 73.12 | 68.01 | 78.13 | 71.71 | 78.52 | 61.39 | 63.75 | 60.11 |
| $N = 4$ | 75.93 | 76.01 | 77.63 | 74.81 | 78.96 | 73.12 | 78.14 | 64.21 | 65.02 | 63.11 |
| $N = 6$ | 76.61 | **76.99** | **77.75** | **74.85** | **80.01** | 75.17 | **78.77** | **65.62** | **65.34** | **63.14** |
| $N = 8$ | **76.73** | 76.51 | 77.32 | 73.92 | 79.69 | **75.41** | 78.72 | 65.59 | 65.31 | 62.98 |

## A.2 ROUGE WITH LN

Considering the compatibility between the LN method and Rouge, we conducted additional experiments by fine-tuning the LayerNorm layer within the blocks while adding RouGE, as shown in 9. The experimental results indicate that adding a small number of parameters, the LN-RouGE model can bring about a slight improvement in accuracy, but it cannot surpass the RouGE model itself entirely. Moreover, adding LN trainable parameters does not reduce the accuracy difference between degraded and non-degraded data groups.

Table 9: Test the combination of LN and the RouGE method.

| Method | Rain200L | | DDN | | GoPro | | LIS | | Snow100k | |
|---|---|---|---|---|---|---|---|---|---|---|
| | Clean | Rainy | Clean | Rainy | Clean | Blurry | Clean | Low-light | Clean | Snowy |
| LN | 75.44 | 74.48 | 72.9 | 69.05 | 78.62 | 71.73 | 78.91 | 65.47 | 64.02 | 62.51 |
| RouGE | 76.61 | 76.99 | **77.75** | **74.85** | 80.01 | 75.17 | 78.77 | 65.62 | 65.34 | 63.14 |
| LN-RouGE | **76.86** | **77.54** | 77.23 | 71.55 | **81.11** | **75.18** | **79.65** | **66.98** | **66.45** | **63.98** |

## A.3 COMPARISON OF USING GLOBAL FEATURE $F_t$ AND LOCAL FEATURE $x_t$

In experiment, we utilize the global feature $F_t$ as a signal to control the gating weights in the probability gate input. The role of Ft is to provide the probability gate with more distinguishable features, thereby reducing the interference of intermediate variables in the model. As a comparison, we conducted additional experiments to compare the effectiveness of using $x_t$ and $F_t$.

Table 10: Performance of RouGE with $x_t$ and $F_t$

| feature | Rain200L | | DDN | | GoPro | | LIS | | Snow100k | |
|---|---|---|---|---|---|---|---|---|---|---|
| | Clean | Rainy | Clean | Rainy | Clean | Blurry | Clean | Low-light | Clean | Snowy |
| $x_t$ | **76.57** | 76.94 | 77.60 | 73.40 | 79.44 | 74.75 | **79.72** | 64.53 | 64.19 | 63.06 |
| $F_t$ | 76.61 | **76.99** | **77.75** | **74.85** | **80.01** | **75.17** | 78.77 | **65.62** | **65.34** | **63.14** |

## A.4 MORE SEGMENTATION RESULT

|  | Clean img. |  | Degraded img |  |
|---|---|---|---|---|
| | w/o RouGE | w RouGE | w/o RouGE | w RouGE |

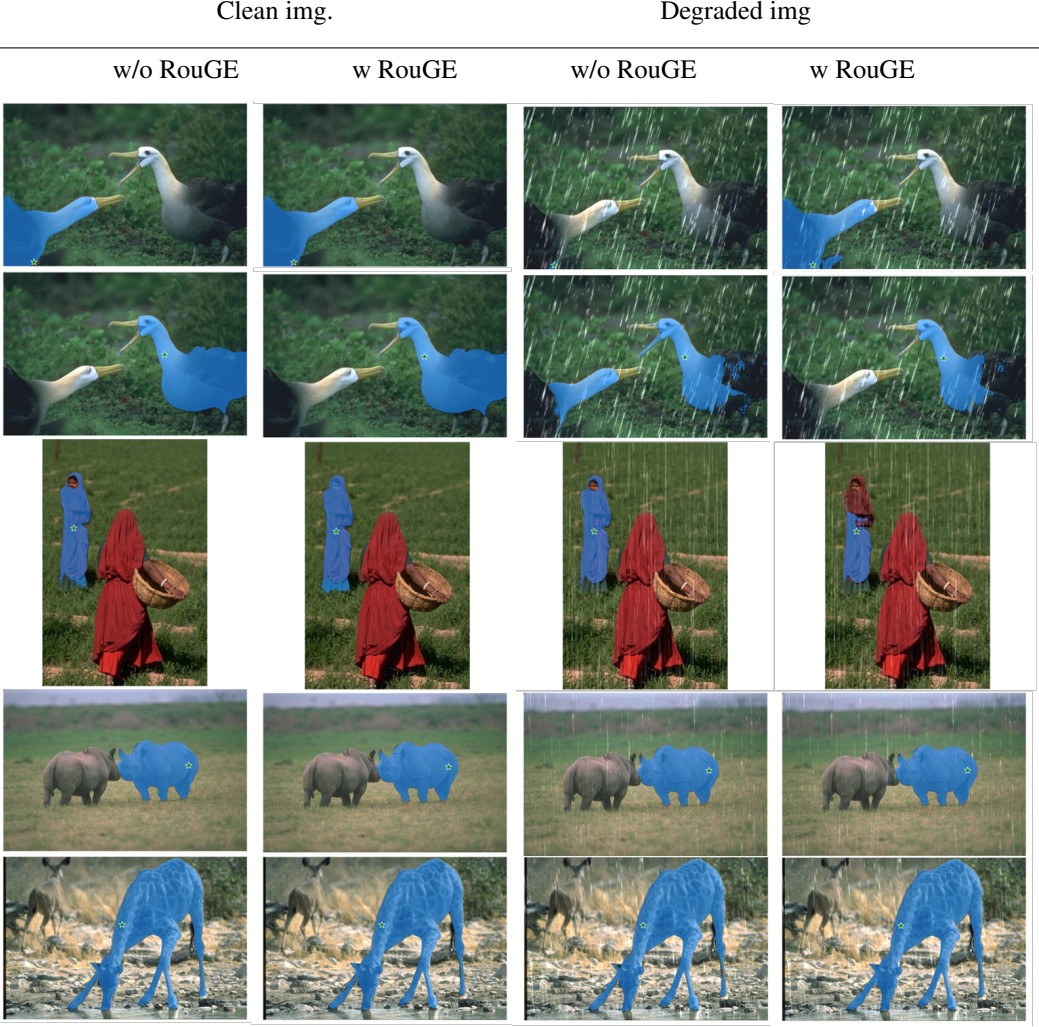

Figure 7: Presentation of the Segmentation performance on Rain200L dataset

Clean img.                                    Degraded img

w/o RouGE          w RouGE          w/o RouGE          w RouGE

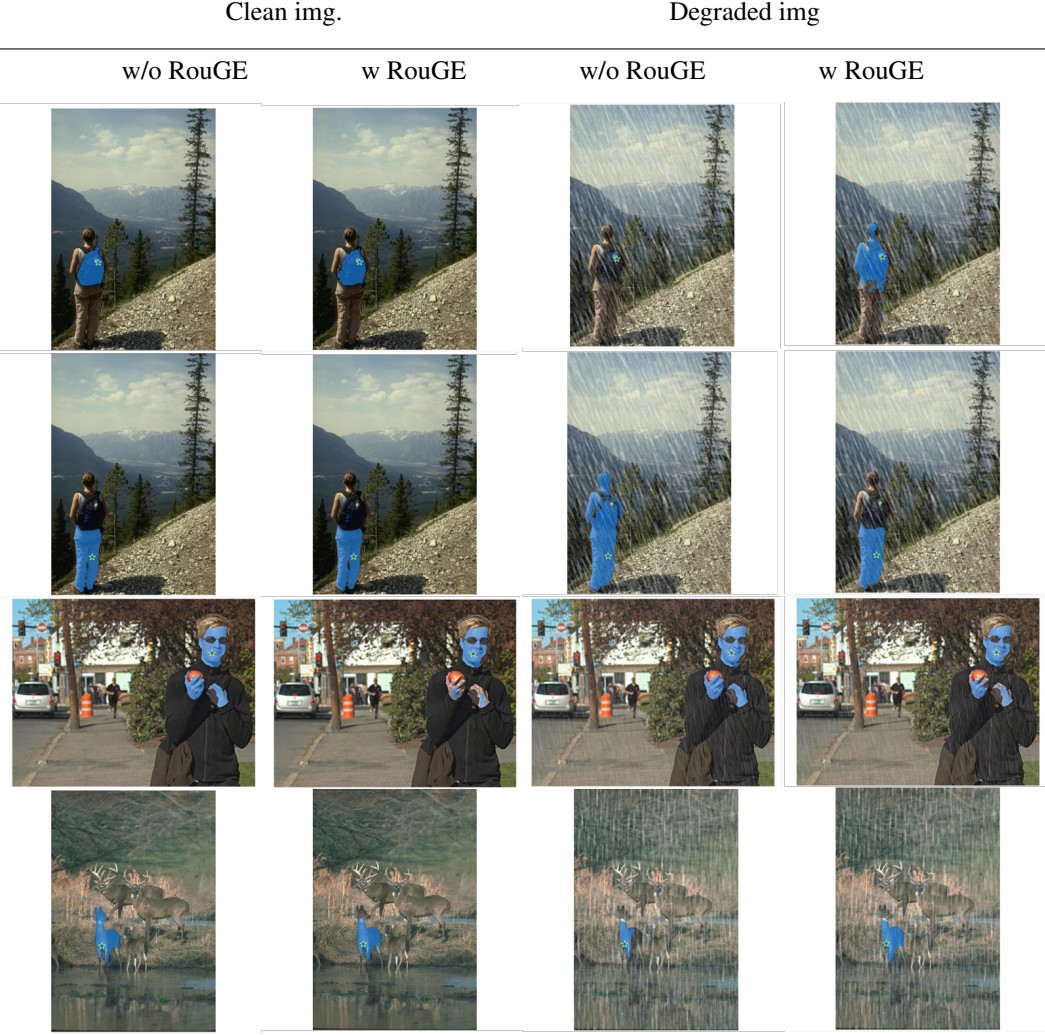

Figure 8: Presentation of the Segmentation performance on DDN dataset

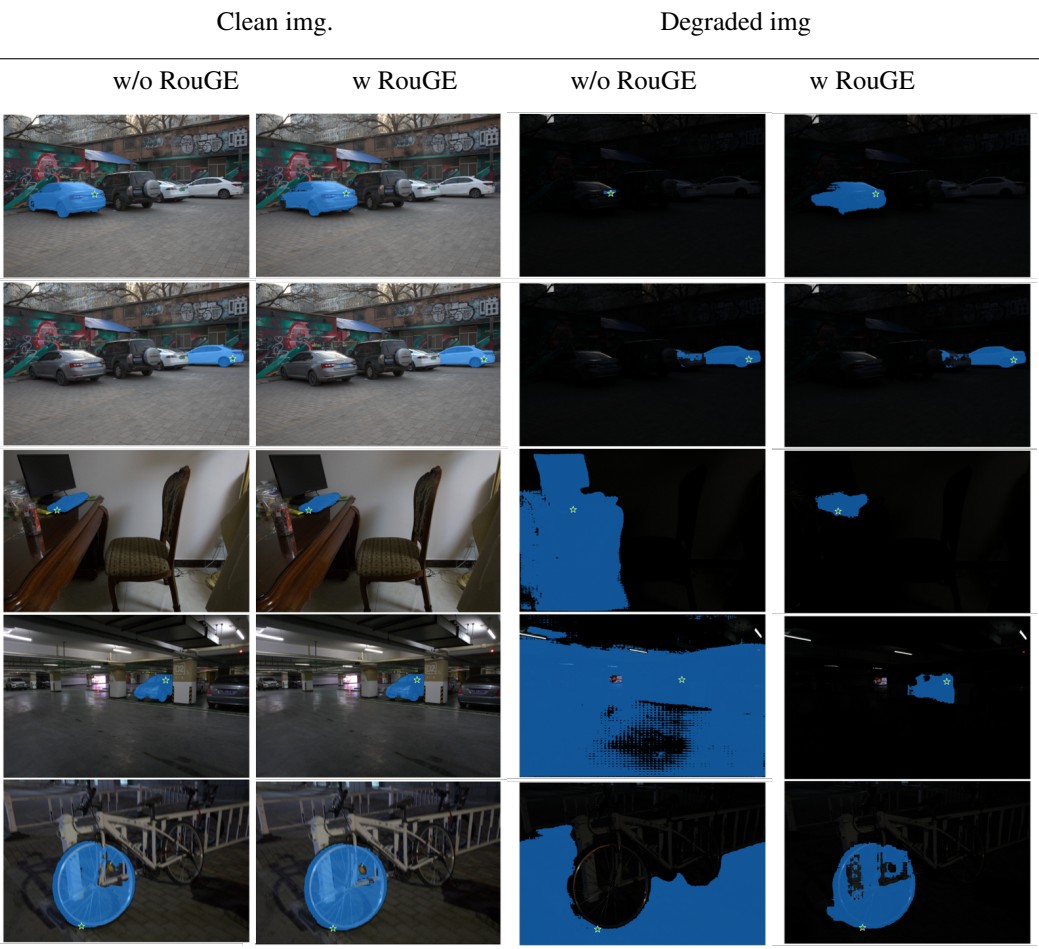

Figure 9: Presentation of the Segmentation performance on LIS dataset

|  | Clean img. | | Degraded img | |
|---|---|---|---|---|
|  | w/o RouGE | w RouGE | w/o RouGE | w RouGE |

Figure 10: Presentation of the Segmentation performance on GoPro dataset

Clean img.                                    Degraded img

w/o RouGE          w RouGE          w/o RouGE          w RouGE

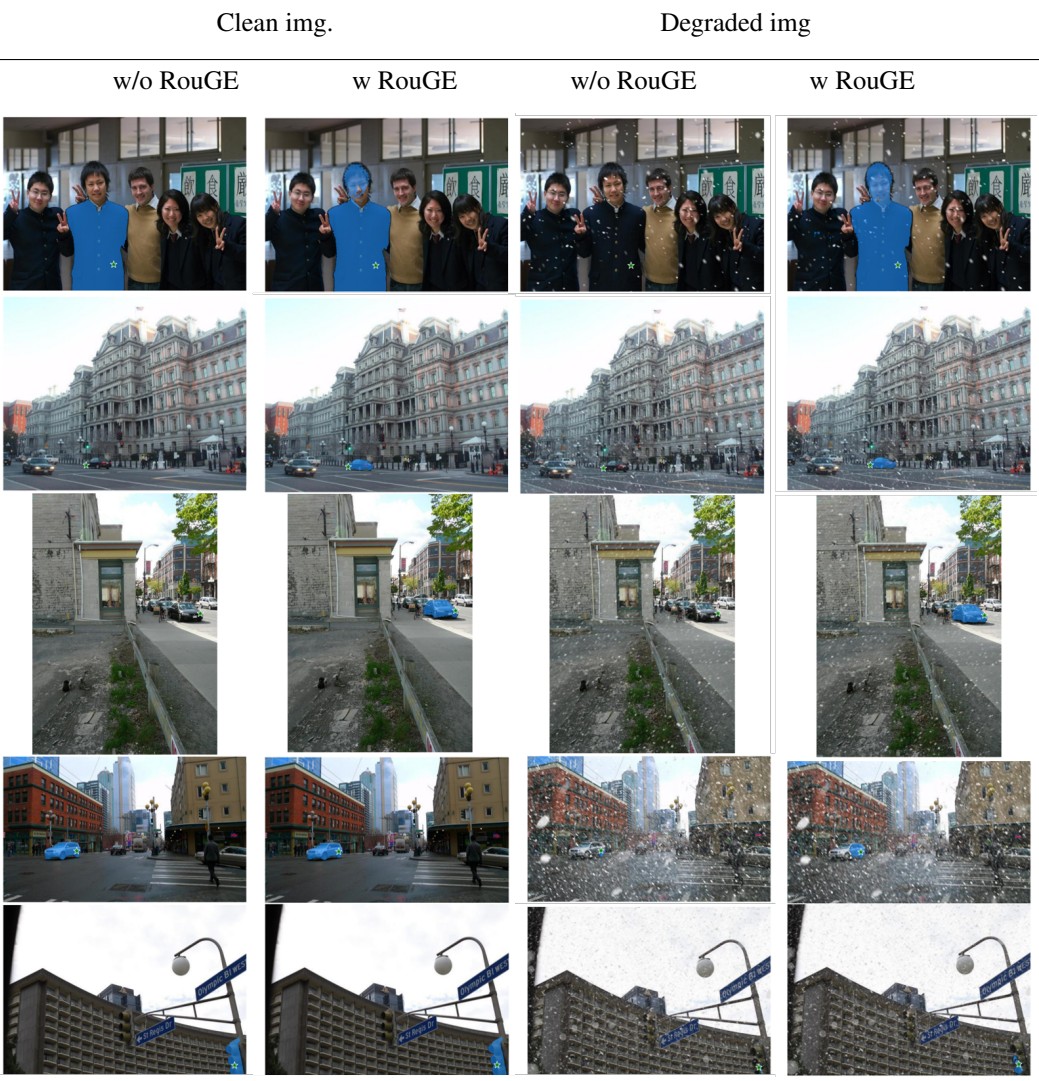

Figure 11: Presentation of the Segmentation performance on Snow100k dataset

