# OpenReview forum: "RouGE: Learning Gated Experts for Segment Anything in the Wild"
_ICLR.cc/2025/Conference — ICLR 2025 Conference Withdrawn Submission_

### Official Review · Reviewer_MspJ · 2024-10-31

**Soundness:** 2
**Presentation:** 2
**Contribution:** 3
**Rating:** 5
**Confidence:** 4

**Summary:**

The paper introduces RouGE, a gated adapter module to enhance the robustness of SAM-based models for degraded images. By employing lightweight probability gates and low-rank experts, RouGE adapts to various degradation types with minimal parameter tuning.

**Strengths:**

- Relevant Motivation: Addressing robustness in SAM-based segmentation models for real-world degraded conditions is a timely and important topic.

- Parameter Efficiency: The use of gated experts and low-rank modules offers an efficient approach to improving robustness with minimal parameter tuning.

- Broad Evaluation: The paper evaluates RouGE on multiple types of degradation and compares it with various parameter-efficient tuning methods, demonstrating its competitive performance.

**Weaknesses:**

1. **Unclear Feature Definition**: The image feature $F_t$ is not clearly defined, leaving ambiguity about its source and role. Please provide a detailed description of the generation process for $F_t$.

2. **Computational Cost of CLIP**: If CLIP is used for $F_t$ extraction, the computational load could be significant. A full breakdown of pipeline steps is needed to assess efficiency.

3. **Unspecified Detection Model**: The object detection model used for bounding boxes is not specified, leaving reproducibility issues and potential variability in results.

4. **Ambiguity in Baseline Comparisons**: The paper uses a sequential adapter integration, whereas Adaptformer has shown that parallel adapter configurations can yield better performance. Without testing RouGE with a parallel MoE adapter setup, the paper does not convincingly demonstrate that its adapter design is optimal.

5. **Missing Adapter Details**: Critical details like the adapter’s bottleneck dimension are omitted, limiting the method's replicability.

6. **Incomplete Ablation Studies**: The paper lacks comprehensive ablation studies for critical hyper-parameters, such as the number of experts and adapter bottleneck dimensions. Including these would strengthen the evaluation and provide insights into the effects of architectural choices on performance.

**Questions:**

Why does the proposed method achieve a lower number of trainable parameters compared to previous methods, despite using a MoE structure? Could the authors provide a quantitative analysis to clarify this?

---

### Official Review · Reviewer_6wJr · 2024-10-31

**Soundness:** 2
**Presentation:** 2
**Contribution:** 3
**Rating:** 5
**Confidence:** 4

**Summary:**

This paper introduces RouGE, a Mixture-of-Experts (MoE) structure designed to enhance the robustness of SAM-based models in handling diverse degraded images, such as low-light, rainy, or blurred conditions. RouGE employs multiple probability gates to assess and adjust features as needed, using low-rank experts to handle different conditions without impairing SAM’s original capabilities. Experiments show that RouGE achieves state-of-the-art performance on both degraded and clean images while tuning only 1.5% of parameters.

**Strengths:**

1. Using a Mixture-of-Experts (MOE) to efficiently enhance the robustness of the foundation model is a sensible idea.

2. The qualitative experiments in both the main text and the appendix are comprehensive, clearly demonstrating the performance improvements of the RouGE model.

**Weaknesses:**

1. Regarding Figures 2 and 3 in the motivation, does the qualitative and quantitative performance degradation of the SAM model occur only on synthesized degraded images? What would the results be if tested on degraded images from the real world? Can the SAM foundation model achieve good results on real-world degraded images?

2. Following up on the first question, would it be possible to add all synthesized degraded images to the SAM foundation model's pretraining dataset to directly improve its performance on degraded images?

3. During training, RouGE uses synthetic image pairs of degraded and clean images. Are the degraded images sourced from the real world or generated? Additionally, how does RouGE perform when real-world degraded images are used during testing?

4. The design of the probability gates lacks novelty. This type of gate, which relies on image features for implicit decision-making, can be applied to various tasks and is not specifically designed for handling different types of degraded images.

5. There is a lack of ablation studies, such as on the number of lazy experts, the number of trainable experts, the adapter design, and the choice of RouGE layer placement, among others.

**Questions:**

please refer to weakness.

---

### Official Review · Reviewer_g9vf · 2024-11-03

**Soundness:** 2
**Presentation:** 1
**Contribution:** 2
**Rating:** 3
**Confidence:** 5

**Summary:**

This paper adapts the SAM-based model for degraded images. The main contribution is on the RouGE layer which has a MoE-like design. The suggested RouGE layer improves the feature modeling by activating the necessary gates. The final model improves the baseline model on different benchmarks.

**Strengths:**

The proposed method can improve the baseline performance and show superior performance compared to other adapt/prompting methods.

**Weaknesses:**

While the improved performance looks promising, the reviewer has several concerns:

1. There are many typos in the current manuscript and the consistency can be improved. For example, the reviewer think in the Figure 1 it should be "Mask" instead of  "Musk"

2. Again in Figure 1, the clean image is the same as the rainy image, while the blurry and low-light ones have different content. This makes the figure hard to understand.

Some more major concerns:

3. It seems that the proposed RouGE has all experts activated, which loses the advantage of boosting inference during inference as in conventional MoE. Therefore, the current method looks more like a combination of different gating functions with a weighted function.

4. It is unclear what experts are activated for which kind of input and how the weighting function varies. In other words, the paper lacks in-depth analysis.

5. The proposed method only compares with one single baseline method. It is unclear if the proposed method can generalize well on the SAM-based works. In addition, the authors replace the proposed adapting/prompting method with other counterparts. While it is so-called a fair comparison, it would be better to at least include the variance of each method. It might be likely that the authors take the best performance from the proposed method and compare it with a lower performance from the other counterparts.

6. From Table 1, it can be seen that LN can already hugely improve the baseline performance with only 10% learnable param of the RouGE block. Therefore, the contribution of the the RouGE block becomes less convincing since we can believe adding more LN might achieve a similar performance. By the way, the proposed RouGE in fact looks very much like a concatenation of LN, with some dimension changes .... So the novelty is limited...

7. The reviewer is not sure if the claim on Real-World degradation is valid since most of the datasets contain only synthetic degradation.,,,

8. There are many more advanced works in the MoE for image restoration as shown in [1]. However, these works are not discussed. The reviewer thinks that the MoE design from this work looks too naive compared to these counterparts...

[1] A Survey on All-in-One Image Restoration: Taxonomy, Evaluation and Future Trends

**Questions:**

See weakness

---

### Official Review · Reviewer_fxk5 · 2024-11-06

**Soundness:** 2
**Presentation:** 3
**Contribution:** 2
**Rating:** 3
**Confidence:** 5

**Summary:**

The paper presents RouGE, a PEFT and MoE framework designed to enhance the robustness of SAM-based models. RouGE employs multiple probability gates to decompose image conditions and manages them using a set of low-rank experts in a PEFT manner. The results show promise results and successfully reducing the number of learnable parameters.

**Strengths:**

1. The idea is simple and the paper is easy to follow.
2. The article's approach shows promising results and robustness improvement on real-world unseen data.

**Weaknesses:**

1. Utilizing MoE to enhance robustness and performance is not a novel concept. The authors integrate MoE with low-rank Adapters in PEFT for the Segment Anything model, which may not provide more insights for the community.
2. Some ablations or results are absent, including Adapter dimensions and a comparison of training sources (e.g., GPU memory and additional computational overhead).
3. Reporting on the performance of structures using MoE with full fine-tuning is essential to evaluate the effectiveness of the PEFT strategy.

**Questions:**

See Weakness. Besides, there are few other suggestions:

1. There seems $2 \times 6 = 12$ Adapter, why does the proposed method have fewer trainable parameters than previous methods like AdaptFormer ?
2. The image feature $F_t$ is not clearly defined.

---

### Note · Authors · 2024-11-13

I have read and agree with the venue's withdrawal policy on behalf of myself and my co-authors.